# Preclinical Models of Adrenocortical Cancer

**DOI:** 10.3390/cancers15112873

**Published:** 2023-05-23

**Authors:** Andrew J. H. Sedlack, Samual J. Hatfield, Suresh Kumar, Yasuhiro Arakawa, Nitin Roper, Nai-Yun Sun, Naris Nilubol, Katja Kiseljak-Vassiliades, Chuong D. Hoang, Emily K. Bergsland, Jonathan M. Hernandez, Yves Pommier, Jaydira del Rivero

**Affiliations:** 1Medical Scientist Training Program, Feinberg School of Medicine, Northwestern University, Chicago, IL 60611, USA; 2Medical Scientist Training Program, Carver College of Medicine, University of Iowa, Iowa City, IA 52242, USA; 3Developmental Therapeutics Branch, National Cancer Institute, NIH, Bethesda, MD 20892, USA; 4Surgical Oncology Program National Cancer Institute, NIH, Bethesda, MD 20892, USA; 5Division of Endocrinology, Metabolism and Diabetes, University of Colorado School of Medicine, Aurora, CO 80016, USA; 6Thoracic Surgery Branch, National Cancer Institute, NIH, Bethesda, MD 20892, USA; 7University of California, San Francisco (UCSF) Helen Diller Family Comprehensive Cancer Center, San Francisco, CA 94158, USA

**Keywords:** genetically engineered model, xenograft, adrenocortical cancer, adrenocortical carcinoma, mouse model, organoid model

## Abstract

**Simple Summary:**

Adrenocortical cancer is a very rare form of endocrine cancer with dismal prognosis. Preclinical models such as cell lines, organoids, and mouse models are essential for both improving basic understanding of this disease and developing treatments. Herein, we review a currently available model for adrenocortical cancers, with a special focus on adrenocortical carcinoma. Recent developments in in vitro models have included cell and 3D culture models with improved recapitulation of the tumor microenvironment and genetics. We hope to improve visibility and access to these models through this review.

**Abstract:**

Adrenocortical cancer is an aggressive endocrine malignancy with an incidence of 0.72 to 1.02 per million people/year, and a very poor prognosis with a five-year survival rate of 22%. As an orphan disease, clinical data are scarce, meaning that drug development and mechanistic research depend especially on preclinical models. While a single human ACC cell line was available for the last three decades, over the last five years, many new in vitro and in vivo preclinical models have been generated. Herein, we review both in vitro (cell lines, spheroids, and organoids) and in vivo (xenograft and genetically engineered mouse) models. Striking leaps have been made in terms of the preclinical models of ACC, and there are now several modern models available publicly and in repositories for research in this area.

## 1. Introduction

Cancers of the adrenal gland, such as adrenocortical carcinoma (ACC), require a unique approach in treatment, diagnosis, and research, owing to their rarity and the multifunctional environment of the adrenal gland. The adrenal gland comprises two embryologically distinct regions: the medulla and the cortex [1]. Within the medulla, chromaffin cells serve the neuroendocrine function of manufacturing epinephrine (adrenaline), a core stimulant of the fight-or-flight response [1,2]. Tumors of chromaffin cells (pheochromocytomas) are often present with classic symptoms of hypertension, headache, and syncope with an incidence rate of 0.04 to 0.95 per 100,000 [3,4,5]. Within the cortex, three distinct zones are present, the zona glomerulosa, zona fasciculata, and zona reticularis; these zones are responsible for manufacturing steroid-based mineralocorticoids, glucocorticoids, and androgens, respectively [6]. Broadly, tumors of the adrenal cortex may be divided into adenomas and carcinomas [7].

Adrenocortical adenoma (ACA) is a benign neoplasm originating from the secretory cells of the adrenal gland, with an incidence of 425 per 100,000 and accounting for 33% to 96% of aberrant adrenal incidentalomas, although this may be an underestimate [8,9,10]. While 85% of ACAs are non-secretory, a subset of ACA may be associated with autonomous glucocorticoid and mineralocorticoid stimulation, and rarely androgen secretion [8,11]. The most common clinical presentation for ACA is autonomous cortisol secretion (Cushing’s syndrome), which presents with symptoms of hypertension, insulin resistance, and obesity, and is associated with increased mortality [7]. Cushing’s occurs in 3.8% to 6.6% of patients with ACA, with a 14.3% risk of developing Cushing’s if the adenoma is >2.4 cm, compared to 3.8% for those <2.4 cm [12,13]. 

ACC is a malignant neoplasm originating from the adrenal cortex [14]. In contrast to the relatively high prevalence of ACA, ACCs may be classified as an ultra-rare disorder, with an incidence of 0.72 to 1.02 per million people/year [15,16,17,18,19]. The prognosis for patients with ACC is poor, with a median 5 year survival rate of 22% and a 10 year survival rate of <5%, accounting for 0.2% of all cancer deaths [14,20]. In 40 to 60 percent of cases, patients with ACC present a chief complaint associated with hormone excess, such as Cushing’s syndrome, gynecomastia, or virilization in women [21,22]. The most common hormonal findings are excess cortisol alone, cortisol paired with androgens, androgens alone, or cortisol paired with an additional cortical hormone, and very rarely mineralocorticoid or estrogen excess [21].

A key identifying feature of ACC which differentiates it from secretory ACA is the identification of secretion from two adrenal zones [7]. ACC is often an aggressive and rapidly proliferative disease. While more than 90% of ACC presents >4 cm, ACC only makes up 10–50% of adrenal masses >4 cm [23,24,25,26,27]. Due to their size, approximately 30% of ACC presents with symptoms associated with tumor mass, such as abdominal or flank pain [23]. The most common sites of metastasis are the liver and lungs, followed by peritoneum, bone, and brain, with half of all patients developing metastasis to multiple organs or sites [28,29]. 

While ACC is rare, there is an increased prevalence of ACC in cancer predisposition syndromes. The Li Fraumeni syndrome (LFS) is a syndrome resulting from an autosomal dominant variant in the *TP53* gene—a gene coding for the tumor suppressor p53—which drastically predisposes toward malignancy [30]. Among patients with LFS, between three and ten percent present with ACC, suggesting a strong influence of germline *TP53* mutation on ACC risk [31,32]. Similarly, the low penetrance alleles of *TP53* have been implicated in a high incidence of childhood ACC, with *TP53* being the underlying genetic cause in 50–80% of childhood ACC [33,34]. In Brazil, a familial low-penetrance (9.9%) *TP53* variant (R337H) contributes to increased ACC development, resulting in the incidence of ACC being nearly 3 to 4 times the global rate at 0.29 to 0.42 per 100,000 [35]. 

In addition to *TP53* variants, ACC is associated with other syndromes. In the case of Beckwith–Weidemann syndrome (BWS)—the most common congenital growth disorder (11p15.5)—resultant tumors may occur within the adrenal cortex, with ACC accounting for 5–15% of tumors in children with BWS [36,37,38]. ACC is also at an increased prevalence in patients with Lynch syndrome, Werner syndrome, and congenital adrenal hyperplasia [39]. Overall, understanding the connection between ACCs and other hereditary diseases has been vital in elucidating the mechanisms of tumorigenesis. However, due to its rarity and poor prognosis, the development of new models of ACC is essential if we are to further explore routes of early diagnosis and therapeutic intervention. 

Current preclinical research models for ACC are limited. There are currently 12 available ACC cell lines (8 human, 4 murine) originating from human ACC (see Table 1) and 6 widely reported primary cultures of human ACC (see Table 2) [40]. Cell culture is often in the form of monolayer or suspension, both of which are relatively cost-efficient and provide relatively reproducible results due to their culture in ideal conditions [41]. Despite their value, neither method adequately replicates the tumor microenvironment (TME). Monolayers have demonstrated impaired cytoskeletal activity and lack cellular diversity, while suspensions do not possess adequate extracellular support [41]. Pre-clinical mouse models of ACC, which more closely capture the tumor microenvironment, have been generated; however, they are also limited. Cell line-derived xenografts (CDXs) are established using cell culture that is subcutaneously administered to immunocompromised mice for localized propagation and in vivo tumor growth [42]. Patient-derived xenografts (PDXs) are generated using tumor fragments from human tissue collected during surgical resection and then directly implanted into immunocompromised mice and are further propagated across generational passages [42]. Both CDXs and PDXs present the ability to observe in vivo human tumor progression; however, there are limitations. CDXs present with atypical histologic findings and poorer chromosomal maintenance when compared to PDXs. PDXs are limited by the supply of viable grafts, complex passaging techniques, and issues associated with clinically relevant dosing [42,43]. There are multiple transgenic mouse models as well, which focus on the manipulation of different growth factors and signaling molecules to produce an ACC-like phenotype. While there are many models present, transgenic mouse models are usually limited to one or two genetic modifications that might only partially recapitulate the heterogeneity of human disease.

This review summarizes the currently used cell line and xenograft models of ACC, which have provided valuable insights into the pathogenesis and natural history of human ACCs. We focused especially on elaborating the details of more in vitro models than current mouse models, although all types of models are included for the completeness of the survey. Newer and more accurate models of ACC are critical to furthering the early detection and targeted treatment of this disease, and a holistic understanding of current models and their advantages and deficiencies is the first step to designing better ones. 

## 2. Results

### 2.1. Cell Lines

Cell lines are summarized in Table 1. Notable primary cultures are summarized in Table 2. The earliest derived ACC cell line for continuous culture was Y-1, initially transplanted from an *Itgal*^−/−^ mouse line, and noted for their continuous production of progesterone derivates from cholesterol [44]. The other major mouse lines, ATC1 and ATC7, were generated from tumors in transgenic mice with SV40-Tag added under the *AKR1B7* promoter. The ATC1 and ATC7 lines have been mostly used in basic endocrine research toward understanding the patterning of and hormonal crosstalk within the adrenal cortex. They have also been used for some therapeutic target discovery, focusing on HOX genes with them [66,67,68].

For decades, the only continuous ACC cell lines available were SW-13 and H295, the latter of which is notable for its sustained steroid secretion even after decades of culture [49,56]. On the other hand, the current consensus is that SW-13, which never produced steroids in culture, is probably derived from a small cell lung cancer metastasis to the adrenal gland, and hence it has fallen out of use in modeling ACC [69,70]. While many more cell lines have been introduced in recent years, H295R remains the most available and heavily used line in current preclinical research [71,72]. In recent years, perhaps not surprising due to its age and passage number, the reproducibility of results across different clones of H295R has been called into question, emphasizing the need not only for new models, but also for avoiding overpassaging models [55]. The ACC HAC15 cell line, first reported in 2008, was later shown to be a subclone of H295R, which had presumably contaminated the attempted culture of a new line [70,73]. Other concerns with H295R cells include their lack of response to ACTH stimulation, which has been attributed to their low expression of its receptor, the melanocortin 2 receptor (MC2R). To remedy this, Nanba et al. used lentiviral particles to introduce the open reading frame of a protein necessary for the surface trafficking of MC2R, the MC2R accessory protein, into H295R cells, generating a strain termed H295RA with the inducible production of 11-deoxycortisol, cortisol, and androstenedione [74].

Excitingly, several more ACC lines have been reported in the last few years, including MUC-1 cell lines in 2016, and CU-ACC1 and CU-ACC2 in 2018 (all three with companion PDX lines for comparison) [55,56]. Recent work has also depended much on primary culture, with more than 40 primary ACC cultures isolated in the last seven years [75,76]. While these cultures have been shared extensively, only the ACC115m clone has been sufficiently immortalized for use as a cell line (now reported as TVBF-7) [40,65]. Hence, H295R remains the only widely available ACC cell line in repositories. Quantitative measures (such as RNA sequencing) of how cell type and function changes in and between models such as tissue culture and PDX vs. in primary tumors are improving, although detailed breakdowns of what changes in particular are present have not yet been developed [56].

Warde et al. recently showed that mitotane sensitivity correlates with intracellular lipid content; while MUC-1 and H295R cells store similar amounts of intracellular lipid droplets, MUC-1 (mitotane resistant) cells are rich in triacylglycerols, whereas H295R (mitotane sensitive) cells are rich in cholesterol esters [77]. Lipid content as measured by Hounsfield units is used in some diagnostic algorithms for ACC; however, these results show that further distinguishing the particular lipids involved may provide more useful clinical information [78,79]. Further investigation of lipid compositions of cell lines may be valuable and can potentially inform the inclusion of such analysis in the future analyses of biopsy and surgical samples to inform the precise treatment of ACC. Recent investigations have also shown crosstalk between adipose stem cells and H295R cells, reinforcing the importance of local lipid metabolism in ACC [80].

### 2.2. Xenografts

While patient-derived xenografts (PDXs) of samples acquired directly from biopsy or surgery into immunodeficient mice are typically recognized as the gold standard for human cancer models, limited PDXs of ACC are available [81,82]. Hence, here, we have included cell line-derived xenograft (CDX) models, which are summarized in Table 3, while PDXs are summarized in Table 4.

The first PDX model was generated from a pediatric patient with ACC and was reported in 2013. No separate cell line of this model has been established [91]. Since then, three new models have been developed, which have companion cell lines [55,56]. Further work has investigated the behavior of one of these models, CU-ACC2-M2B, in a humanized mouse model to better understand the efficacy of checkpoint inhibitor immunotherapy [56,93].

Modern ACC PDX lines not only retain significant molecular similarity (as confirmed by IHC) to their primaries, but also recapitulate the differences between those primaries and some of the heterogeneity of the disease [56,93].

### 2.3. 3D Models

Refer to Table 5: Two primary ACC 3D models (one spheroid, one organoid) have been reportedly recently [94,95]. Prior to 2022, 3D models of ACC consisted only of spheroids generated from H295R and SW-13 cells, primarily used in drug-screening protocols [96,97]. One additional H295R-derived spheroid model was also developed last year [98]. As with cell lines, newer models are increasingly moving toward larger-scale biobank models that will enhance the heterogeneity of models available for future research, although these models are not yet publicly available [95]. 

In addition, a transwell model of ACC co-culture with adipose stem cells showed evidence of crosstalk and worsened disease phenotype induced by the adipose stem cells [80]. Beyond the lipid microenvironment specifics of endocrine cells, co-culture experiments are increasingly important for understanding metastasis and immune response or lack thereof, the latter of which is critical for better improving immunotherapy outcomes. 

3D models are particularly promising in the complex microenvironment of the adrenal cortex as an opportunity to better recapitulate tissue zonation. Recent experiments have also looked at interactions between ATC7 cells and human monocytes, showing that activation of intra-adrenal immune cells may play a role in stimulating steroidogenesis or proliferation [66]. 

A 2022 work by Bornstein et al. on standardized 3D culture techniques has yielded promising results in both replicating H295R and MUC-1 data and establishing additional primary cultures of ACC successfully. Bornstein et al. also worked with bovine and porcine adrenal organoids, but this work was focused primarily on normal tissue working toward transplantation rather than disease. Notably, this comparative work on porcine and bovine organoids also made progress toward the co-culture of medullary and cortical tissue [94].

Although not yet peer reviewed, Dedhia et al. released promising organoid models of ACC, studying metastasis through matrix metalloproteinase experiments in organoids and microfluidic models [99].

### 2.4. Genetically Engineered Mouse Models

While this review will not go into extensive detail about current mouse models, they are summarized here for completeness, and presented in Table 6. A more thorough review which particularly focused on them was recently published by Basham et al. [100]. Relatively many models have been developed to understand adrenocortical neoplasia as opposed to other neuroendocrine, as summarized in Table 5 [101]. Early models mostly focused on the role of *IGF2* [102,103]. While it has been confirmed to be involved in the development and progression of tumors, it is no longer seen as likely to be a driver of oncogenesis itself [104,105,106]. Although no longer a central focus of transgenic models, a study continues on elucidating the mechanism of *IGF2*’s role in adrenocortical neoplasia. Pereira et al. showed that its effects on H295R cells could preferentially be inhibited by mTOR pathway inhibition vs. MEK/MAPK/ERK pathway inhibition [103].

Other recent work has focused more on *CTNNB1*, *APC*, *WNT*, *ZNRF3,* and *TP53* [107]. Val and coauthors recently showed evidence that phagocytic macrophages may be involved in the relatively higher prevalence of ACC in women via a conditional *ZNRF3* KO model [108].

**Table 6 cancers-15-02873-t006:** Genetically engineered mouse models of neuroendocrine neoplasia.

Model Name	Type ^1^	Type ^2^	Gene (Promoter)	Year	Reference
P540scc-SV40	ACC	transgenic	*SV40-TAg* (*CYP11A1*)	1994	[109]
FG-Tag	NEPC, ACT	transgenic	*SV40-TAg* (*HBG*)	1996	[110,111,112]
PEPCK-IGF-II	ACC	transgenic	*IGF2* (*PEPCK*)	1999	[102]
Nr5a1^+/−^	ACT	KO	*NR5A1*	2000	[113]
AdTAg	ACC	transgenic	*SV40-TAg* (*AKR1B7*)	2000	[54,114,115]
YAC-TR	ACT	transgenic	*NR5A1* (YAC)	2007	[116]
FAdE-SF1	Pediatric ACT	transgenic	*NR5A1* (FAdE)	2009	[117]
ACD^acd/acd^::p53^+/−^	ACC	KO	*ACD*, *TP53*	2009	[118]
APC KO	ACC	TS KO	*APC*	2012	[104]
Adlgf2	ACC	transgenic	*IGF2* (*AKR1B7*)	2012	[105]
H19ADMD	ACC	transgenic	*APC*, *IGF2/H19-ICR* (*NR5A1*)	2012	[104]
Apc^+/−^	ACC	KO	*APC*	2014	[119]
RNF43^−/−^	-	TS KO	*RNF43*	2019	[120]
ZNRF^−/−^ CTNNB1^+/−^	adrenal hyperplasia	TS KO	*ZNRF3, CTNNB1*	2019	[120]
ZNRF^−/−^	-	TS KO	*ZNRF3*	2019	[120]
p53-LOF (AS*^Cre/+^*::Trp53*^flox/flox^*), PCre^AS/+^	-	TS transgenic	*TP53*	2020	[121]
βcat-GOF (AS*^Cre/+^*::Ctnnb*^flox(ex3)/+^*), BCre^AS/+^	-	TS transgenic	*CTNNB1*	2020	[121]
p53-LOF/βcat-GOF (AS^Cre/+^::Trp53*^flox/flox^*::Ctnnb*^flox(ex3)/+^*), BPCre^AS/+^	ACC	TS transgenic	*TP53, CTNNB1*	2020	[121]
Znrf3*^flox/flox^* SF1-Cre^high^	ACC	TS KO	*ZNRF3*	2022	[108]

^1^ ACC: adrenocortical carcinoma, NEPC: neuroendocrine prostate cancer. ^2^ KO: Knockout, TS: Tissue-specific.

## 3. Discussion

In comparison with ACC explants, several features are important to consider, including genetics, hormone secretion, and growth patterns. We summarize below a synthesis of the processes used for verifying the ACC115m primary culture, associated TVBF-7 cell line, and Bornstein et al.’s spheroid models as a paradigm for an appropriate analysis and confirmation of samples [55,58,94].

To verify the authenticity of primary cultures, cell lines, and xenografts, it is valuable to perform short tandem repeat (STR) profiling in comparison to primary samples. Numerous cases of contamination across cell lines and the overgrowth of lymphocytes or other cells instead of intended tumor cells have reinforced the necessity of such verification.

In characterizing any sample ACC cells, exome sequencing of at least driver- (*TP53*, *MEN1*, *PRKAR1A*, *CTNNB1*, *APC*, *ZNRF3*, *IGF2*, *EGFR*, *RB1*, *BRCA1*, *BRCA2*, *RET*, *GNAS* and *PTEN*) and steroidogenesis-related (*CYP11A1*, *CYP17A1*, *HSD3B2*, *HSD17B4*, *CYP21A2*, *CYP11B1*, *CYP11B2*, *MC2R*, *AT1R*) genes should be performed, if not more comprehensive sequencing. Hormone secretion of cortisol, aldosterone, dehydroepiandrosterone, dehydroepiandrosterone sulfate, testosterone, and 17-hydroxyprogesterone should ideally also be screened by mass spectrometry. For primary tissue samples, xenografts, and organoids, immunostaining should be performed for Ki-67 and for the endocrine-specific markers *SF1*, *EGFR*, and 3β-hydroxysteroid dehydrogenase.

It is also important for appropriate positive and negative controls to be used in analyzing the secretion and stimulation of endocrine cells, as many popular forms of media, such as Nu-serum, contain hormones such as testosterone [122]. Researchers should ensure that their measurements compare to an appropriate baseline (i.e., of complete media before culture of cells) and use appropriate controls.

In trying to convert less aggressive phenotypes to lines suitable for in vitro study, transgenic models with such genes as SV40-TAg are often used, and ACC models such as ATC1 and ATC7 use this technique. However, we urge caution with such approaches as they may no longer resemble their original less aggressive phenotype. Instead, we encourage more complex culture models that better recapitulate the original environment, such as the standardized spheroid model of Bornstein et al. reported above or other 3D systems. Such systems are also valuable in analyzing the co-culture of ACC with other cell types such as adipose cells or lymphocytes, which are essential to understanding the lipid and immune microenvironment of ACC. As standardized 3D culture systems become a reality, ideally, co-culture techniques will also become more refined and widespread in understanding ACC.

## 4. Conclusions

Adrenocortical carcinoma is an aggressive orphan malignancy with limited therapeutic options. Its rarity has slowed clinical research advances. As a result, preclinical models are doubly important in understanding ACC’s pathogenesis and potential treatment. Since the development of the first ACC model systems (mouse and human cell lines) in the 1960s and 1970s, much of the research has focused on the use of those now widely available systems. However, many novel systems of various types have been developed since. In particular, biobanking and standardized protocols have led to the generation of more patient-derived models in recent years. Unfortunately, only a few of these have reached wider usage and public availability such as in biobanks and mouse repositories. While there is a critical demand for new ACC model systems, it is just as important that existing model systems are shared and cross-validated across different research groups and between one another. 

Using a variety of models is essential to capture the heterogeneity of clinical disease and to compensate for the flaws that different model systems have. No single model system can perfectly recapitulate disease, but the use of multiple models with complementary strengths will bring us closer to that understanding. Moving fast in research is sometimes essential, and simpler in vitro systems (such as monolayer and spheroid culture) perform this admirably. Slower and more relevant in vivo data using mouse or other xenograft models provide some information about how ACC interacts with the rest of the body, but an orthotopic model would be better for showing these interactions than existing flank models. More complicated in vitro systems that incorporate the 3D organization of cells or larger organoid structures can help to bridge the gap between the former simpler and the latter more complex models. In addition, human ACC tumor-bearing immunocompromised mice are useful for the exploration of potential therapeutic approaches, including chemotherapy, targeted therapy, or a combination of these modalities. To accelerate ACC immunotherapy, there is also an urgent need to develop rapid syngeneic mouse models rather than genetically engineered mouse models with slow tumor development.

The concerted application of existing models and the development of new ones to fill gaps will improve preclinical understanding and empower future clinical research on ACC. In particular, the major gaps in current ACC preclinical models are a comparison across newer model systems and the development of better in vitro model systems for organoid or more complex 3D cultures.

## Figures and Tables

**Table 1 cancers-15-02873-t001:** Adrenocortical carcinoma-derived cell lines.

Study	Cell Line	Source	Repository	Ref. No. ^1^	Notes	Year	Reference
Yasumura, 1966	Y-1	mouse	ATCC	48	Produces 20α- and 11β-20α-hydroxyprogesterone in culture, cannot produce corticosterone due to lack of CYP21 expression [44,45].	1966	[46]
Gazdar, 1990	H295R	primary	ATCC ^2^	717	Has an activating S45P *CTNNB1* mutation [47,48]. Limited response to ACTH stimulation, although compensatory variant has been generated (see text).	1980	[49]
Rahman, 2001	Calpha1	mouse	-	1	Generated by introduction of SV40-TAg expression under *INHA* promoter, ACC developed only by mice which were gonadectomized prepubertally. Limited use.	2001	[50]
Schteingart, 2001	RL-251	primary	-	6	Limited response to ACTH stimulation. Secretion of IL-8 and angiogenic factors.	2001	[51]
Ueno, 2001	ACT-1	primary	-	1	Expression of 3β-hydroxysteroid dehydrogenase. Significant chromosomal abnormalities, modal number 61.	2001	[52]
Ragazzon, 2004	ATC1	mouse	-	8	Generated by introduction of SV40-Tag expression under *AKR1B7* promoter. Zona fasciculata phenotype. ACTH responsive, corticosterone production positive.	2004	[53]
Ragazzon, 2006	ATC7	mouse	-	8	As ATC1.	2006	[54]
Hantel, 2016	MUC-1	PDX	-	33	Nuclear expression of *SF1*, cytoplasmic expression of 3β-hydroxysteroid dehydrogenase. Cortisol production positive.	2016	[55]
Kiseljak-Vassiliades, 2018	CU-ACC1	PDX	-	6	G34R *CTNNB1* mutation. Cortisol and corticosterone production positive, aldosterone production negative (although primary tumor was aldosterone-secreting, metastases from which line is derived were not). ACTH unresponsive.	2018	[56]
Kiseljak-Vassiliades, 2018	CU-ACC2	PDX	-	6	G245S *TP53* mutation. ACTH unresponsive. Minor cortisol secretion. Deletion of *MSH2* exons 1–6.	2018	[56]
Landwehr, 2021	JIL-2266	primary	-	1	Hemizygous mutations in *MUTYH* and *TP53*. Insignificant hormone secretion. High mutational burden relative to most ACC.	2021	[57]
Sigala, 2022	TVBF-7	primary	-	2	Q247* *APC* mutation (nonsense). Derived from primary culture ACC115m (Table 2). Significant expression of *MC2R* compared to H295R, but limited responsiveness to ACTH stimulation.	2022	[58]

^1^ Determined as number of non-reviewed, non-conference references on PubMed or PubMed Central. ^2^ Parent cell line NCI-H295 is only available from BCRC; H295R is the most widely used subclone.

**Table 2 cancers-15-02873-t002:** Adrenocortical carcinoma-derived primary cultures.

Study	Culture	Source	Repository	Ref. No. ^1^	Notes	Year	Reference
Almeida, 2008	Almeida pediatric	primary	-	1	Survived to eight passages.	2008	[59]
França, 2013	ACC-T36	primary	-	6	Demonstrated that forced expression of *TCF21* reduced expression of *SF1.*	2013	[60]
Gara, 2015	BD140A	primary	-	7	Generated at Phoenix Translational Genomics Research Institute, limited information available in publication.	2015	[61,62]
Fragni, 2019	Fragni series	primary	-	2	Six unique reported cultures.	2019	[63]
Abate, 2020	ACC24-I	primary	-	1	Metastasis derived. Previously treated with EDP + M.	2020	[64]
Rossini, 2021	ACC115m	primary	-	3	Lymph node metastasis derived. Primary non-secretory. Survived continuous culture as TVBF-7 cell line (Table 1).	2021	[65]

^1^ Determined as number of non-reviewed, non-conference references on PubMed or PubMed Central.

**Table 3 cancers-15-02873-t003:** Cell line-derived xenografts.

CDX Line	Host	Source	Notes	Year	Reference
RL-251 (Schteingart, 2001)	SCID	RL-251	Seeded cells produced detectable circulating IL-8 and ENA-78 in xenografted mice.	2001	[51]
Doghman, 2010	NOD/SCID/γ_c_^null^	H295R	Showed miRNAs miR-99a and miR-100 coordinately regulate expression of mTOR in ACC.	2010	[83]
Doghman, 2012	NOD/SCID/γ_c_^null^	H295R	Showed dual inhibitor of PI3K/mTOR reduced ACC xenograft growth.	2012	[84]
Doghman, 2013	NOD/SCID/γ_c_^null^	H295R	Showed that mitotane does not inhibit the growth of H295R xenografts long-term even with sustained therapeutic levels.	2013	[85]
Nagy, 2015	BALB/c SCID	H295R	Showed mitotane inhibits xenografted tumor growth.	2015	[86]
Hantel, 2016	NMRI nu/nu	H295R	Identified TNFAIP3/A20 overexpression as mechanism of TNFα inhibition resistance in xenografted ACC.	2016	[87]
Nilubol, 2018	Nuþ/Nuþ	H295R, BD140A, SW-13	Showed combination of flavopiridol and carfilzomib inhibits xenografted tumor growth.	2018	[62]
Cerquetti, 2019	nu/nu Forkhead mice	H295R, SW13	Showed radiosensitizing effect of mitotane to inhibit tumor growth in full-body irradiation of xenografted mice.	2019	[88]
Nadella, 2020	nu/nu	H295R	Showed c-KIT inhibitor inhibits xenografted tumor growth.	2020	[89]
Laha, 2022	Nuþ/Nuþ	H295R, SW-13	High-throughput drug screening identifying combination inhibition of MELK and CDK as potential therapeutic target.	2022	[71]

**Table 4 cancers-15-02873-t004:** Primary tumor-derived xenografts.

Study	PDX Line	Host	Notes	Year	Reference
Pinto, 2013	SJ-ACC3	CB17 scid^−/−^	First (pediatric) ACC PDX. Primary-derived. Treatment naïve at establishment. Successfully reseeded into multiple different lines [90].	2013	[91]
Hantel, 2016	MUC-1	NMRI nu/nu	First adult ACC PDX. Neck metastasis derived; originally treated with EDP + M ^1^.	2016	[55]
Kiseljak-Vassiliades, 2018	CU-ACC1	nu/nu	Perinephric metastasis derived. Treatment naïve at establishment. Androgen-secreting primary.	2018	[56]
Kiseljak-Vassiliades, 2018	CU-ACC2	nu/nu	Liver metastasis derived, post-mitotane, SBRT ^2^, and embolization.	2018	[56]
Kar, 2019	CU-ACC9	nu/nu	Primary-derived. Cortisol-secreting primary. Originally treated with EDP + M ^1^.	2019	[92]
Lang, 2020	CU-ACC2-M2B	BRGS	CU-ACC2 variant in humanized mouse model for immunotherapy studies.	2020	[93]

^1^ EDP + M: Etoposide, doxorubicin, cisplatin, and mitotane. ^2^ SBRT: Stereotactic body radiation therapy.

**Table 5 cancers-15-02873-t005:** Adrenocortical carcinoma-derived 3D models.

3D Model Line	Type	Source	Notes	Year	Reference
Nilubol, 2012	spheroid	H295R, SW3	Bortezomib, ouabain, methotrexate, and pyrimethamine showed inhibitory activity against spheroids and monolayers.	2012	[96]
Armignacco, 2019	transwell	H295R	Co-culture of H295R monolayer above an adipose stem cell monolayer leads to reprogramming of both cell types, leading to more aggressive disease phenotype.	2019	[80]
Cerquetti, 2021	spheroid	H295R	Sorafenib inhibited growth and caused disaggregation of tumor spheroids.	2021	[8]
Fudulu, 2021	transwell	ATC7	Co-culture of ATC7 monolayer below human monocytes to study immune interactions and cross-talk, showing that IL-6 release by monocytes may modulate steroidogenesis.	2021	[66]
Langer, 2022	spheroid	H295R	Albumin-stabilized carrier nanoparticles efficiently delivered higher doses of mitotane to spheroids than are possible in aqueous solution.	2022	[98]
ACC15m (Bornstein, 2022)	spheroid	primary	Lymph node met-derived. Treatment history unpublished. No hormone secretion.	2022	[94]
Laha, 2022	aggregate	H295R, SW-3	High-throughput drug screening identifying combination inhibition of MELK and CDK as potential therapeutic target.	2022	[71]
Bornstein, 2022	spheroid	primary, MUC-, H295R	Standardized spheroid generation in a custom 24-well plate format. Proof-of-concept for generation and pharmacological testing of not only malignant but also benign hyperplasia derived spheroids.	2022	[94]
Baregamian, 2023	organoid	primary	Three ACC and five benign adrenal neoplasia established in continuous organoid culture. Hormone secretion reduced after second passage.	2023	[95]

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
