# Peer review of "Preclinical Models of Adrenocortical Cancer"

_cancers, 2023, doi:10.3390/cancers15112873_

Round 1

Reviewer 1 Report

The paper "Preclinical models of adrenocortical cancer" is a valuable document that surely readers will want to keep for reference guidance, although it could have gone much further.

In fact most of the references to cell lines, organoids, xenographs and genetically modified mice that have been used as preclinical models for the study of adrenocortical cancer are just enumerated and not really discussed.

A few points that may be corrected: In the introduction - "Increased mortality" is not a symptom. It´s a consequence.

LiFraumeni syndrome is not "an autosomal variant in the TP53 gene" It's a syndrome resulting from such variant...

resultant Wilms tumors may occur within the adrenal cortex ??

In Table 1, in the legend, it is said "H295R is most widely used child". The sentence is surely incomplete, unfinished!

In Table 2 - I believe that instead of "PDX line" it should be "CDX line"

In Table 4 - I was surprised that there is only 1 citation of an "organoid" study

In section 2.4 when speaking  of the use of "many models developed to understand adrenocortical neoplasia" it is said that early models mostly focused on the role of IGF2. The authors cite only the study of Weber et al which was done in Transgenic mice and should pehaps also cite a recent study involving the most commonly used cell line (IGF2 role in adrenocortical carcinoma biology.   doi: 10.1007/s12020-019-02033-5)

I have no particular comments on the quality of the English language. There are some sentences that are clearly mistaken but I have alluded to them in the comments for Authors

Author Response

We thank the reviewer for their kind and helpful feedback. Several major additions have been made in response to the cumulative notes by the reviewers. In particular:

  1. "Increased mortality" on line 56 has been rephrased.
  2. Reference to Li Fraumeni syndrome on line 78 has been clarified.
  3. Erroneous reference to adrenal Wilms tumors in the adrenal gland has been corrected on line 89.
  4. Table 1 legend has been corrected.
  5. Table 3 (formerly 2) column 1 title has been corrected to CDX line.
  6. Reference and information from Weber et al has been added in section 2.4.
  7. We have added reference in the text to one more recent organoid model available in preprint only currently. To our knowledge, these are the only two presently published models of organoids in ACC, an area which these authors agree should be expanded upon.
  8. We have added additional context about current work IGF2 and reference to Pereira et al.

Thank you!

Reviewer 2 Report

In the current manuscript entitled ‘Preclinical models of adrenocortical cancer 2’, Sedlack and colleagues provide a thoughtful and well-referenced review that does a good job at assembling a list of cell lines, organoids and mouse models while highlighting their importance for the study of ACC. This review brings the readers up-to-date publications and could be an essential resource for scientists conducting research in this area. I only have minor comments and maybe suggest missing information to be added if the authors agree.

1. It would be helpful for the readers and anyone interested in generating new adrenal cell lines or organoids to add some references to protocols used to dissociate human or mouse adrenal tissues. Are there specific conditions to consider when developing a new adrenal cell line? Are growth factors or signaling molecules particularly helpful for successfully establishing a new cell line? Standardized protocols are mentioned in the discussion, but no references are associated.

2. Regarding Table 1, the reviewer wondered if a column could be added to mention important specificities of each cell line. For example, Y1 cell lines do not express CYP21 and, therefore cannot produce corticosterone (Parker et al. 1985, Szyf et al., 1990) but metabolize cholesterol to 11b-20a-dihydroxyprogesterone (Pierson et al., 1967) due to abnormally high 20alpha keto reductase activity. Another example, H295R harbors a CTNNB1 mutation (S45P), leading to the constitutive transcriptional activity of β-catenin-LEF/TCF (Tissier et al., 2005). They also present with low expression of MC2R and consequently are not or minimally responsive to ACTH stimulation. An additional cell line, H295RA, has been generated to restore ACTH sensitivity (Nanba et al., 2016). TVBF-7 line harbors nonsense mutation for APC (Sigala et al., 2022), CU-ACC1 cells have a mutation in CTNNB1, CU-ACC2 cells a TP53 mutation, and loss of MSH2 (Kiseljak-Vassiliades et al., 2018) etc… These characteristics are essential when considering using cell line to test pathway or influence of chemical compounds on steroidogenic activity.

3. Differences in cell lines also include intracellular lipid content. For example, Warde et al. 2022 demonstrated that H295R preferentially store cholesteryl esters while MUC-1 store triacylglycerol. The first one is mitotane-sensitive, while the second is mitotane-resistant.

4. The reviewer was wondering why the first authors are not mentioned in Table 1 similar to what was provided in Table 2. The university of origin could also be added for a rapid way to identify cell lines.

5. The reviewer believes that some cell lines are possibly missing, such as ATC7 & ATC1 (Ragazzon et al, 2004 and 2006) and Calpha-1 cells (Rahman et al., 2001).

6. What does TS stand for in TS KO. There are a few acronyms not detailed (SBRT, LN..) across the manuscript.

7. The paragraph on the 3D is very short, could the authors expand on that, maybe comment on the ‘zonation’ of 3D culture?

8. In table 5, the AdTag are mentioned twice

9. Could the authors include a paragraph with recommendations/guidelines when generating a new cell line.. Could the authors comment on the main characteristics to assess? For example, the abstract of the paper establishing the TVBF-7 recapitulated most of them to the reviewer’s knowledge?

Mutational status of important driver genes (TP53, MEN1, PRKAR1A, CTNNB1, APC, ZNRF-3, IGF-2, EGFR, RB1, BRCA1, BRCA2, RET, GNAS and PTEN), Wnt-signaling specificities (CTNNB1 mutation vs. APC mutation vs. wildtype), steroidogenic-(CYP11A1, CYP17A1, HSD3B2, HSD17B4, CYP21A2, CYP11B1, CYP11B2, MC2R, AT1R) and nuclear-receptor-signaling (AR, ER, GCR), hormone secretion profiles (Cortisol, Aldosterone, DHEA, DHEAS, Testosterone, 17-OH Progesterone, among others) under basal and stimulated conditions’

10. Strajhar et al., 2017 mentioned the presence of testosterone in Nu-serum, thus possibly leading to misinterpretation when assessing the steroid capacity of new cell lines; this information could be helpful for readers when considering using cell lines as a preclinical model.

11. Given the presence and possibly the importance of resident and recruited immune cells in the development of adrenal diseases (Dolfi et al., 2022, Wilmouth et al., 2023), could the author comment on possibilities to better mimic the microenvironment in the future, implementing co-culture for example?

12. Line 156. The reviewer is not sure about the meaning of the sentence: ‘Eight and thirty two primary ACC cell lines…’

Author Response

We thank the reviewer for their kind and helpful comments. Significant content has been added throughout the paper on the basis of theirs and the other reviewers feedback, in addition to the specific points below.

  1. We have added content on best practices for establishment and characterization of models in the discussion.
  2. Useful details about cell lines and primary cultures have been added.
  3. A paragraph discussing application of lipid content has been added, however, due to lack of analysis of it in many cell lines it has not been added as a column to the table.
  4. First authors were added to tables 1 (and new table 2), however, universities of origin were left off due to space constraints.
  5. Calpha1, ATC1, and ATC7 cells have been added, thank you for pointing out their absence!
  6. TS (tissue-specific), SBRT (stereotactic body radiation), and LN (lymph node) have been footnoted when presented in tables or defined parenthetically at their first appearance in the text.
  7. Information about 3D models that seek to capture more of the zonation process has been added to the 3D models section.
  8. The duplicate reference to AdTag mice has been removed.
  9. See 1.
  10. This information has been added to the discussion section (per 1).
  11. Recap of existing co-culture models such as those with adipose stem cells has been added, and discussion of future directions (in particular with respect to immune interactions) has been added to the discussion.
  12. This line has been amended for clarity.

Thank you!

Reviewer 3 Report

Sedlack et al. report in their manuscript “Preclinical models of adrenocortical cancer” on preclinical ACC models (mouse and human derived cell culture models, genetically engineered mouse models, PDX and CDX xenografts). The topic is interesting as it is currently a highly dynamic field with many encouraging developments and the authors give a nice and broad overview on different model types. The written result section is very brief, but I think together with the provided tables including relevant references it gives a compact overview. However, I have some minor and major concerns:

The study equally lists primary cultures and cell lines. Primary cultures are since many years now widely in use in laboratories working on adrenal tumors, in specific on ACC. However, for many years the development of human ACC cell lines was not successful. Thus, please clearly distinguish between these two model types.

Up to my knowledge the currently broadly accepted cell lines of mouse origin are Y1, ATC1 and ATC7. The two latter ones have been established, characterized and published in 2004 and are currently lacking in the overview.

Adrenocorticotropin-dependent changes in SF-1/DAX-1 ratio influence steroidogenic genes expression in a novel model of glucocorticoid-producing adrenocortical cell lines derived from targeted tumorigenesis.

Ragazzon B, Lefrançois-Martinez AM, Val P, Sahut-Barnola I, Tournaire C, Chambon C, Gachancard-Bouya JL, Begue RJ, Veyssière G, Martinez A. Endocrinology. 2006 Apr;147(4):1805-18.

Regarding human cell lines NCI-H295, MUC-1, CU-ACC1, CU-ACC2, JIL-2266 and TVBF-7 have been reported and characterized in detail as cell lines including patient histories, confirmation of repeated and ongoing passaging/immortality, investigation of appropriate ACC markers and by providing distinct STR-profiles which clearly distinguish the new cell lines from NCI-H295 and thereby also confirm that a contamination with NCI-H295 cells (as happened before in two labs) is thereby excluded. By checking the references provided here I can not follow why other models are listed in the same way as the described cell lines. For example ACC-T36 and BD140A seem to be ACC primary cultures, “Almeida pediatric” is included in the JCEMs scientific abstract as primary culture of an adenoma, “the Fragni series” were clearly reported as primary cultures and not as cell lines as they also did not show continuous growth. A few additional infos in this direction are already provided below the table, but I would propose to make this difference absolutely clear as it can lead otherwise to confusion about existing, available and widely accepted human cell lines of ACC. As mentioned before primary cultures are widely used in labs worldwide and are important tools. Thus, if the authors would like to give a few examples please clearly include these as own sub-column in the table including the appropriate info. There is one exception: ACC115m has been first reported as primary culture and upon continuous in vitro growth a subsequent detailed characterization followed including all necessary characteristics mentioned above. Afterwards, ACC115m has been defined as cell line and re-named to TVBF-7 (as reported in Ref 60 and reviewed in 40).

Thus, please overwork the lines 96/97, table 1 and paragraph 2.1 in this regard and clearly distinguish between cell lines and primary cultures. Moreover, please focus in the paragraph “Cell lines” more on the existing and better characterized full panel of ACC cell lines including NCI-H295, MUC-1, CU-ACC1, CU-ACC2, JIL-2266 and TVBF-7. HAC-15 represent a sub-clone of NCI-H295, but is not an own cell line as mentioned in line 152 and SW-13s origin is not clear to be ACC. In fact, up to my knowledge in the original publication also given here as reference 45 it is definitively described as metastasis of a small cell carcinoma in the adrenal gland. Thus, I would propose to exclude SW-13 as nowadays equally accepted ACC model or to further strengthen the critical introduction currently included.

Line 98 – up to my knowledge it is a wrong reference, this review does not sum up the overall numbers listed, but discusses the main human cell lines mentioned above.

Table 2 / PDX Lines: the author write the treatment history is unpublished, but the history is provided in the original article added here as reference 54 (section “establishment and characterization of MUC-1 xenografts”). Please correct.

Table 4 ACC115m up to my knowledge all treatment histories are given in the supplements. It is again given in the section “Establishment of a Novel Cancer Cell Line, TVBF-7” in Ref Ref 60. Please add. In the same paper as here cited for ACC115m, a wide range of 3D models has been reported including also 3D models for MUC-1 and NCI-H295. Please complete and also add MUC-1 to 2.3.

Sedlack et al. report in their manuscript “Preclinical models of adrenocortical cancer” on preclinical ACC models (mouse and human derived cell culture models, genetically engineered mouse models, PDX and CDX xenografts). The topic is interesting as it is currently a highly dynamic field with many encouraging developments and the authors give a nice and broad overview on different model types. The written result section is very brief, but I think together with the provided tables including relevant references it gives a compact overview. However, I have some minor and major concerns:

The study equally lists primary cultures and cell lines. Primary cultures are since many years now widely in use in laboratories working on adrenal tumors, in specific on ACC. However, for many years the development of human ACC cell lines was not successful. Thus, please clearly distinguish between these two model types.

Up to my knowledge the currently broadly accepted cell lines of mouse origin are Y1, ATC1 and ATC7. The two latter ones have been established, characterized and published in 2004 and are currently lacking in the overview.

Adrenocorticotropin-dependent changes in SF-1/DAX-1 ratio influence steroidogenic genes expression in a novel model of glucocorticoid-producing adrenocortical cell lines derived from targeted tumorigenesis.

Ragazzon B, Lefrançois-Martinez AM, Val P, Sahut-Barnola I, Tournaire C, Chambon C, Gachancard-Bouya JL, Begue RJ, Veyssière G, Martinez A. Endocrinology. 2006 Apr;147(4):1805-18.

Regarding human cell lines NCI-H295, MUC-1, CU-ACC1, CU-ACC2, JIL-2266 and TVBF-7 have been reported and characterized in detail as cell lines including patient histories, confirmation of repeated and ongoing passaging/immortality, investigation of appropriate ACC markers and by providing distinct STR-profiles which clearly distinguish the new cell lines from NCI-H295 and thereby also confirm that a contamination with NCI-H295 cells (as happened before in two labs) is thereby excluded. By checking the references provided here I can not follow why other models are listed in the same way as the described cell lines. For example ACC-T36 and BD140A seem to be ACC primary cultures, “Almeida pediatric” is included in the JCEMs scientific abstract as primary culture of an adenoma, “the Fragni series” were clearly reported as primary cultures and not as cell lines as they also did not show continuous growth. A few additional infos in this direction are already provided below the table, but I would propose to make this difference absolutely clear as it can lead otherwise to confusion about existing, available and widely accepted human cell lines of ACC. As mentioned before primary cultures are widely used in labs worldwide and are important tools. Thus, if the authors would like to give a few examples please clearly include these as own sub-column in the table including the appropriate info. There is one exception: ACC115m has been first reported as primary culture and upon continuous in vitro growth a subsequent detailed characterization followed including all necessary characteristics mentioned above. Afterwards, ACC115m has been defined as cell line and re-named to TVBF-7 (as reported in Ref 60 and reviewed in 40).

Thus, please overwork the lines 96/97, table 1 and paragraph 2.1 in this regard and clearly distinguish between cell lines and primary cultures. Moreover, please focus in the paragraph “Cell lines” more on the existing and better characterized full panel of ACC cell lines including NCI-H295, MUC-1, CU-ACC1, CU-ACC2, JIL-2266 and TVBF-7. HAC-15 represent a sub-clone of NCI-H295, but is not an own cell line as mentioned in line 152 and SW-13s origin is not clear to be ACC. In fact, up to my knowledge in the original publication also given here as reference 45 it is definitively described as metastasis of a small cell carcinoma in the adrenal gland. Thus, I would propose to exclude SW-13 as nowadays equally accepted ACC model or to further strengthen the critical introduction currently included.

Line 98 – up to my knowledge it is a wrong reference, this review does not sum up the overall numbers listed, but discusses the main human cell lines mentioned above.

Table 2 / PDX Lines: the author write the treatment history is unpublished, but the history is provided in the original article added here as reference 54 (section “establishment and characterization of MUC-1 xenografts”). Please correct.

Table 4 ACC115m up to my knowledge all treatment histories are given in the supplements. It is again given in the section “Establishment of a Novel Cancer Cell Line, TVBF-7” in Ref Ref 60. Please add. In the same paper as here cited for ACC115m, a wide range of 3D models has been reported including also 3D models for MUC-1 and NCI-H295. Please complete and also add MUC-1 to 2.3.

Author Response

We thank the reviewer for their kind and helpful feedback. Significant additions and modifications have been made throughout the manuscript in response to all three reviewers comments, but in particular:

  1. We have separated the table of cells in use into a table of immortalized cell lines (table 1) and a table of primary cultures (table 2). 
  2. In addition to NCI-H295, MUC-1, CU-ACC1, CU-ACC2, JIL-2266 and TVBF-7, we have also retained the reported cell lines ACT-1, RL-251, and SW-13 in this table. Although their origins and characterization are less definite than the prior lines, they are included here for completeness, and the uncertainty is now noted in the text. The incorrectly reported subclone of H-295R, HAC15, which was previously included in the table for completeness although noted to not be a unique cell line in the footnotes, is now only reference in the text.
  3. Information about the mouse lines ATC1 and ATC7 has also been added.
  4. Treatment history for ACC115m has been added.
  5. The 3D models discussed in Bornstein 2022 have been added.

Thank you!

Round 2

Reviewer 3 Report

The manuscript improved significantly, my congratulations for this comprehensive overview!